# The Quality of Life in Patients with Epilepsy in the Context of Sleep Problems and Sexual Satisfaction

**DOI:** 10.3390/brainsci11060778

**Published:** 2021-06-11

**Authors:** Hanna Rozenek, Kornelia Zaręba, Jolanta Banasiewicz, Stanisław Wójtowicz, Tomasz Krasuski, Krzysztof Owczarek

**Affiliations:** 1Department of Medical Psychology and Medical Communication, Medical University of Warsaw, 02-091 Warsaw, Poland; hanna.rozenek@wum.edu.pl (H.R.); jolantabanasiewicz.wum@gmail.com (J.B.); stwojt@o2.pl (S.W.); tomasz.krasuski@wum.edu.pl (T.K.); krzysztof.owczarek@wum.edu.pl (K.O.); 2First Department of Obstetrics and Gynecology, Center of Postgraduate Medical Education, 01-004 Warsaw, Poland

**Keywords:** sleep problems in epilepsy, quality of life in epilepsy, sexuality in epilepsy

## Abstract

Epilepsy influences the quality of patient functioning in almost all aspects of life. This study aimed to assess the quality of difficulties with sleep initiation and those concerning sexual function, and to assess whether sleep disorders are associated with selected aspects of sexual function and quality of life. The study involved completing a quality-of-life questionnaire for persons with epilepsy: a QOLIE-31 questionnaire, and the present authors’ questionnaire, including 20 questions. A total of 170 questionnaires were completed by 89 men and 81 women. Sleep problems of various frequency were reported by 98 participants (57.6%). Only 41% of patients were definitely satisfied with their sex life. The respondents who declared dissatisfaction with their sex life more often reported difficulties with sleep initiation (χ2 = 10.20; *p* ≤ 0.001). Moreover, those who claimed that epilepsy might contribute to sex life problems more commonly declared dissatisfaction with their sex life (χ2 = 20.01; *p* ≤ 0.001) and more frequently, reported difficulties with sleep initiation (χ2 = 6.30; *p* ≤ 0.012). The issues should constitute the constant element of medical history taking, as improvement in these areas of functioning may promote better quality of life in patients with epilepsy.

## 1. Introduction

The term “quality of life” is used in medical science in the context of health- and non-health-related consequences of diseases. It is considered an important measurement of health [1]. Human quality of life is determined by numerous objective (external influence) and subjective factors. The extrinsic factors include financial situation, level of education, place of residence, professional activity, and family and social relationships. The intrinsic factors are associated with the perception of situations in which human life undergoes considerable changes. They are determined by the personality shaped by previous life experiences. The perception of one’s own quality of life is also affected by temperament, which is mainly determined by genetic factors [2,3].

Quality of life is self-assessed as being poorer by patients with epilepsy because they cope with numerous problems in everyday life, which influence their quality of life [2,3]. The underlying causes are psychosocial issues, mood disorders, medications, and epileptic seizures. Moreover, they are frequently associated with sleep initiation and sleep disorders, which may contribute to poorer control of seizure episodes [4]. A study conducted in Australia indicated that the quality of life in patients with epilepsy is influenced by age, income, the frequency of seizures, number of antiseizure drugs and perceived prosperity [5]. Interestingly, elderly people (> 65 years old) achieved higher scores in the quality-of-life assessment (QUOLIE-31 questionnaire). Sleep initiation difficulties may result in frustration, dissatisfaction, irritability and the perception of one’s sexual attractiveness, both in men and women. The facts directly translate into emotion-related factors, determining the quality of life. Falling asleep is frequently associated with persistently waiting for sleep to come. At the same time, one may experience sadness, depressive thoughts, anxiety and fear. The reminiscence of everyday events results in their critical judgment. Traumatic situations, disgrace and failures are recollected over and over. Therefore, it is not surprising that a depressive and anxious attitude toward reality may be shaped as a result [6,7]. Stauder et al. reported that 18% of respondents with epilepsy stated they had suicidal ideation [7]. Moreover, the syndrome of the following, gloomy day, which results from sleepiness, influences the quality of life in its numerous aspects. Research conducted in China included adults with epilepsy and demonstrated that symptoms related to depression and fear corresponded with the deterioration of the subjective quality of sleep and insomnia [8]. Polytherapy and professional activity also contributed to increased daytime somnolence in patients with epilepsy [9]. Issues concerning the psychosocial problems of patients with epilepsy were addressed in other publications [2,3,10,11]. Undoubtedly, according to the literature, this group of patients often coped with various challenges, generating stress and tension [12]. Stress-related tension may impede sleep initiation and affect sleep quality. CRH (cortico-liberin) plays an important role in the physiology of a stress reaction. It initiates the release of glucocorticoids and is a neurotransmitter that activates all pathways associated with fear, anxiety and brain arousal. It impairs sleep via a direct effect on the brain and, partially, because it activates the sympathetic nervous system. Stress hormones are characterized by the negative influence on cognitive processes. They are destructive for endocrine metabolism, e.g., by contributing to the reduction in the levels of reproductive hormones [13]. Interestingly, Merz conducted research and presented conclusions in which the role of reproductive hormones was emphasized both in relation to stress and memory [13]. Undoubtedly, a cause-and-effect correlation is present between the tension resulting from stress, sleep problems and issues with sexual functioning. Patients may experience frustration due to difficulties in the sexual aspect, which may cause sleep problems, which, in turn, may trigger sexual function issues [14].

The publication presented by a team of Norwegian scientists concentrated around Complex Epilepsy Research Group in Oslo University Hospital emphasized that a satisfying sex life plays an important role not only in the assessment of psychosocial functioning and maintaining health, but also in the process of restoring health and treatment [15]. The significance of the sexual activity of humans is not only related to the biological aspect, but also to preserving family and interpersonal relationships. It promotes high self-esteem and provides positive emotions, such as the sense of accomplishment and happiness. The frequency of sexual dysfunction in persons aged over 40 is 40%, while in those aged over 70, it amounts to 70%. The most common dysfunctions are the loss of libido in women (22%) and premature ejaculation in men (21%). Erectile disorders of various degrees are experienced by 52% of men aged 40–70 with 2/3 being moderate-degree disorders. Sexual dysfunctions impair the quality of life not only in people suffering from mental disorders, but also in healthy individuals [16]. 

Regrettably, sexual dysfunctions are frequently reported by patients with epilepsy. Henning et al. demonstrated that women with epilepsy more commonly experienced orgasmic disorders and vaginal dryness, while men more commonly developed erectile problems followed by reduced sex drive and premature ejaculation. The studied patients also had fewer sexual partners over the year preceding the study [17]. According to clinicians sexual difunctions occurred more commonly in patients with epilepsy than in those with other chronic neurological pathologies. However, research describing sexual behaviors in patients with epilepsy is scarce [18,19]. Moreover sexual disorders occur more commonly in persons with more frequent episodes, focal epilepsy, those who receive enzyme-inducing antiseizure drugs and using polytherapy, and with drug-resistant epilepsy [20,21].

The authors of scientific papers emphasized the common mechanism of sleep disturbance in the context of sexual disorders, which was explained by such factors as the influence of androgens on sexual functions. It is considered that those hormones play an important part in male and female sexuality (e.g., via the influence of the level of libido, arousal and the number of sexual fantasies experienced daily). Androgens also influence the duration and quality of sleep in both sexes [22]. Poor quality of sleep may be associated with the frequency of seizures, daytime somnolence and depression [23]. Furthermore, sleep disorders are considerably more common in patients with drug-resistant epilepsy and focal epilepsy [24,25]. Problems with sleep initiation are undoubtedly related to the deterioration of the quality of life in individuals with chronic diseases and in healthy ones. They are relatively common in patients with epilepsy and occur twice as frequently in patients with partial seizures in the course of epilepsy [23]. Chronic insomnia symptoms and short sleep duration are associated with the deterioration of mental health and poorer quality of life in patients with epilepsy [26].

Undoubtedly, the search for factors that demonstrate both the qualitative and quantitative correlation with the quality of life in this group of patients still constitutes a considerable challenge for clinicians who view patients, using a holistic approach. Therefore, the improvement of the quality of life of patients with epilepsy is a valid aim of their treatment [4]. Thus, we focused on patients’ opinions regarding selected aspects of their functioning in the sexual aspect and the context of sleep quality. The following research questions were proposed:Do the respondents report problems with sleep initiation?Are the respondents satisfied with their sex life?Do the respondents think that epilepsy contributes to problems with their sex life?What is the correlation between the declared answers (questions 1–3) concerning selected aspects of sexual function and sleep?Does the assessment of the quality of life of patients differ between the following groups: those who declare satisfaction vs. no satisfaction with their sex life; those who consider epilepsy to be the cause of sex life problems vs. those who express the opposite opinion; and those who declare sleep initiation problems vs. no sleep initiation problems?

## 2. Materials and Methods

### 2.1. Material

The exploratory study was conducted in accordance with the Declaration of Helsinki, and the protocol was approved by the Ethics Committee of the Medical University of Warsaw (KB/146/2008). The study group included men and women with epilepsy. The inclusion criterion was the patients’ age being over 18. The oldest patient was 84 years old. Furthermore, the authors excluded patients with a new diagnosis of epilepsy, due to the fact that quality of life assessment might be affected by emotional reactions resulting from the early phase of adaptation to the disease. The minimal time that passed between diagnosis and taking part in the study was one year. The study material was collected between the years 2008 and 2011 as a part of the research into the quality of life of patients with epilepsy in selected health care facilities in Poland, where patients with epilepsy were treated, i.e., Bydgoszcz, Ciborz, Lublin, Kielce, Warsaw and Szczecin.

The patients were treated in neurology departments or they attended appointments in neurology clinics. Participation in the study was voluntary. The patients were enrolled after their informed consent was obtained. A total of 200 patients with the diagnosis of epilepsy were asked to participate in the study. A total of 170 individuals (89 men and 81 women) completed the questionnaire, which constituted a 75% response rate. Regrettably, not all items were completed correctly, which was included in the results.

### 2.2. Method

#### 2.2.1. QOLIE-31 Questionnaire

The authors of the study used the QOLIE-31 questionnaire, which contains 31 questions and is used in the assessment of the quality of life of patients with epilepsy. In the course of the psychometric analyses, the following 7 factors were distinguished in the questionnaire: SW—seizure worry, OQ—overall quality of life, EWB—emotional well-being, EF—energy and fatigue issues, COG—cognitive functioning, ME—medication effects, and SF—social functioning. The decision to use the QOLIE-31 questionnaire was made based on the fact that it is a simple tool that does not pose problems for the patients regarding its completion. At the same time, the thematic range facilitates the investigation of numerous significant aspects of the quality of life in patients with epilepsy. It comprises the categories and standards associated with the quality and level of life in patients with epilepsy, which were specified by the special commissions of the International League Against Epilepsy and the International Bureau for Epilepsy [10]. Furthermore, the average values of Cronbach’s alpha obtained in a study conducted by Owczarek and Michalak [27] were similar to the average values of this coefficient obtained in a study conducted in the U.S.A. by Cramer et al. in a validation dataset [11]. This indicates the analogous reliability of the questionnaire obtained in the Polish population of patients with epilepsy. We obtained the authors’ permission for the use of this tool.

#### 2.2.2. The Authors’ Own Questionnaire

The questionnaire consisted of two parts and included 20 questions (Appendix A). Part I (4 questions) was addressed by the attending physicians to obtain information concerning the clinical status of the patient. It tackled aspects such as age, sex, duration of the disease, the etiology of epilepsy, medications used, drug-resistance, the presence of an epileptic seizure, and type and frequency of seizure episodes. Part II of the questionnaire (16 questions) was completed by the patients and included questions concerning the influence of the disease on the patients’ lives. Therefore, it referred to the individual aspects of the patients’ functioning, which might affect the perceived quality of life. This part of the questionnaire also included the following questions, referring to sexual function and sleep:(1)Are you satisfied with your sex life?(2)Do you have problems with falling asleep?(3)Do you think epilepsy contributes to problems with sex life?

The QOLIE-31 and the present authors’ questionnaire were an integral part of the interviews conducted with patients, being an element of the follow-up visits in neurology offices and clinics. The present study included only three questions from the questionnaire to differentiate between the groups (satisfaction with sex life, problems with sleep initiation, and declaration that epilepsy contributed to difficulties in sex life).

### 2.3. Statistics

In order to meet the aims of the present paper and answer the research questions, the responses were divided with reference to the individual questions and classified into two groups for each question:-Responses not satisfied vs. satisfied with sexual life: Those selected on the scale from “often” to “always” were treated as positive (YES), while those from “rarely” to “never” were negative (NO).-Respondents having vs. not having problems with sleep initiation: Responses selected on the scale from “sometimes” to “always” were treated as positive (YES), while those from “rarely” to “never” were negative (NO).-Respondents declaring that epilepsy was vs. was not the cause of difficulties in their sex life: Responses selected on the scale from “rarely” to “always” were treated as positive (YES), while those from “very rarely” to “never” were negative (NO).

The assignment of the responses to two categories—YES and NO declarations—was due to the distribution of the results and the search for an optimal solution to demonstrate the differentiating effect.

The Kolmogorov–Smirnov test was used in the analysis. It showed that the data met the criteria of normal distribution. The comparative analysis of results concerning the quality of life, assessed with the QOLIE-31 questionnaire in the distinguished groups of patients, was performed with a Student’s t-test. The criterion of statistical inference was set at the level of significance of *p* < 0.05. The analyses were conducted with IBM SPSS Statistics for Windows, Version 24.0., Armonk, NY: IBM Corp. (released 2016).

## 3. Results

We obtained a total of 170 questionnaires completed by 89 men and 81 women. The average age of the participants was 39.57 years (SD = 13.70). In some cases, we did not succeed in collecting complete data. Therefore, the results refer to the number of respondents (N) who provided data.

### 3.1. Sociodemographic Data

Most respondents, 52 (30.8%), lived in villages. The remaining participants lived in towns, 27 (16.0%), and in big towns and cities (Table 1).

A total of 82 (48.5%) out of 169 respondents declared to be in a relationship. Information concerning the marital status of the group is included in Table 1. Among 163 individuals who answered the question concerning the offspring, 80 respondents (48.2%) had no children, 41 (24.7%) had one child and 45 (27.1%) had more than one child (Table 1). The highest percentage of the respondents, 84 (49.4%), reached the secondary level of education. Only 20 (11.8%) patients had primary education (Table 1).

No possible interactions were demonstrated between variables such as age, sex, marital status, which might be important both in terms of the subjective opinions of the participants (regarding sleep disorders and satisfaction with sex life) and for the assessment of the quality of life. 

### 3.2. Medical History Data

The etiology of epilepsy was unspecified in almost half of the cases 76 (44.7%). The most common types of seizures were generalized tonic-clonic ones and complex partial seizures. However, the majority of respondents had not experienced them for the previous 6 months. The occurrence of a seizure (since the previous visit) was reported in 7 patients, which constituted 4.1% of the sample. The physicians stated that during the treatment period, epilepsy was drug-resistant in 85 (50%) respondents. Monotherapy was used in 76 patients (44.7%), while polytherapy was used in 91 (53.5%) patients. Table 2 includes information concerning the type and frequency of seizures in the study group.

### 3.3. The Frequency of Sleep Initiation Problems

Sleep problems of various frequency were reported by 98 participants (57.6%) (Figure 1). The problems never occurred or occurred very rarely in 72 individuals (42.4%). According to the abovementioned classification, YES (sometimes, often, very often, always) was declared by 75 individuals, while NO (seldom, very seldom, never) by 95.

### 3.4. Satisfaction with Sex Life

A total of 69 patients (41%) were definitely satisfied with their sex life, as they answered “always” or “very often”. Over 35 persons (20.8%) replied “does not apply”. This answer was significantly more common in elderly participants (Figure 1). The response “does not apply” was not included in any category of answers because it is difficult to understand it as satisfaction or dissatisfaction with sex life. According to the abovementioned classification, YES (often, very often, always) was declared by 76 individuals, while NO (sometimes, seldom, very seldom, never) by 57.

Only 54 participants claimed that epilepsy contributed to problems with sex life. The majority of them expressed the opposite opinion: “very seldom” was declared by 81 (49.7%) participants. Moreover, we obtained the following responses: “always (5), “seldom” (9), “often” (4) and “very often” (17). Only 28 (17.2%) participants answered “never” and 19 answered “sometimes” (Figure 1). According to the abovementioned classification, YES (seldom, sometimes, often, very often, always) was declared by 54 individuals, while NO (very rarely, never) by 109.

Additionally, the analysis of the results with the use of the χ2 test showed that the respondents who declared dissatisfaction with their sex life more often reported sleep initiation problems (χ2 = 10.20; *p* ≤ 0.001). Moreover, those who claimed that epilepsy might contribute to sex life problems more commonly declared dissatisfaction with their sex life (χ2 = 20, 01; *p* ≤ 0.001) and more frequently reported sleep initiation problems (χ2 = 6, 30; *p* ≤ 0.012).

### 3.5. The Quality of Life in the Context of the Quality of Sleep and Sex Life

The respondents who declared no sleep initiation problems were characterized by a significantly better quality of life in all its aspects, except the EF (energy and fatigue issues) scale problems associated with fatigue and the lack of energy, which did not reach the level of statistical significance (Table 3).

The respondents who declared satisfaction with sex life were characterized by better general assessment of the quality of life, compared to those who were dissatisfied with their sex life. The trend was also visible for the following scales: SW (seizure worry), EF (energy/fatigue), COG (cognitive functioning), and SF (social functioning). No significant differences were reported for other aspects (Table 4).

The respondents who claimed that epilepsy was the cause of difficulties in sex life assessed their quality of life as being poorer compared to those who expressed the opposite opinion. The correlation was observed both regarding the general assessment of the quality of life and all its subscales (Table 5).

## 4. Discussion

The difference between the two groups in which the participants declared vs. did not declare sleep initiation problems was statistically significant (t = −5.18; *p* ≤ 0.001) regarding the total score obtained in the QOLIE-31 questionnaire. Patients who reported no problems with sleep initiation obtained higher scores with regard to quality of life, compared to those who declared such problems. Therefore, difficulties with sleep initiation constituted an important aspect of the everyday functioning of patients with epilepsy and influenced the parameters of the quality of life. Quigg et al. conducted a study with the use of the Insomnia Severity Index (ISI) and demonstrated that insomnia is a common problem in individuals with epilepsy and, similar to our study, is associated with poorer quality of life [28,29]. Problems with sleep initiation, declared in the present study, influenced almost all the subscales of the QOLIE-31 questionnaire. As regards the factors contributing to the quality of life, we observed statistically significant differences between both groups referring to all subscales of the questionnaire, except the EF scale (energy and fatigue problems) for which the differences were statistically insignificant. Similar results were obtained in a study conducted by Mexican authors [25]. They also noted that such problems were underestimated and constituted an important aspect of the daily functioning of patients who suffered from various diseases, not only epilepsy. Obviously, sleep problems and sex life satisfaction vary among people of different ages. Therefore, we performed new analyses, limiting the age of the participants to the 19–68 range. The results were similar. 

The only scale that remained unaffected by sleep initiation problems in the present study was the EF scale (energy and fatigue problems). Notably, sleep is not a uniform and unchangeable phenomenon. It is possible that in the present study group, the total amount of sleep was sufficient for physical rest but in the qualitative aspect, it was characterized by the reduced consolidation of memories, which normally occurs during deep sleep. It may be indicated by the results obtained in the COG (cognitive function) subscale. Another justification of this empirical fact may be related to the procedure of data collection in the present study. The results may have been affected by such factors as the time of measurement. This factor was not controlled but it may be assumed that the respondents might differ in terms of the level of energy throughout the day. The effects of insufficient sleep manifesting as energy distortions at the behavioral level and perceived exhaustion may be overlooked in the morning. It might underlie the fact that patients with epilepsy did not report problems of fatigue. The effects of insufficient sleep are usually experienced on the subsequent day. They may intensify throughout the day and be much more apparent late in the afternoon and in the evening. Notably, the patients might have also attempted to cope with the consequences of a sleepless night. It was emphasized in the context of individuals with insomnia, who implemented various strategies to deal with its effects: sleeping longer in the morning and napping during the daytime [30].

The difference between two groups in which the participants declared satisfaction or dissatisfaction with sex life appeared to be statistically significant (t = 3.08; *p* ≤ 0.003) regarding the total score obtained in the QOLIE-31 questionnaire. The respondents who declared satisfaction with their sex life were characterized by better general quality of life compared to those who were dissatisfied with sex life. Similarly, statistically significant differences were observed regarding the following subscales: SW—seizure worry, EF—energy and fatigue problems, COG—cognitive functioning, and SF—social functioning. Sex life is one of the elements that is disrupted as a result of the use of antiepileptic drugs [31]. Our results showed that the assessment of the erotic aspect of functioning was closely associated with other areas of life. It results from the fact that sexual activity is important for people in terms of achieving the following aims: meeting the biological need to reproduce, strengthening interpersonal relationships, and satisfying the feeling of being needed in family and society [32]. Sexual satisfaction promotes higher self-esteem (it offers the possibility of finding fulfillment in the sexual role) or experiencing positive emotions of contentment, and happiness. Individuals satisfied with their sex life may perceive their quality of life to be better. They may also be less concentrated on the negative consequences of the disease and its treatment. In the present study, 35 respondents (20.8%) replied “does not apply” to the question concerning sex life. The group may have included elderly people. However, it may also have been stated by other patients who declared epilepsy to be an obstacle, so they do not undertake this kind of activity.

Analyses conducted in the present study revealed that patients who declared dissatisfaction with their sex life more commonly reported problems with sleep initiation. This may be related to the fact that sex life problems develop due to various etiologies. Apart from organic causes, common underlying factors may be purely psychological or socio-cultural [33]. As regards epilepsy, it is worth noting its specificity is caused by such factors as a variety of manifestations and their frequency [34,35]. It is also one of the stigmatized diseases [6]. 

Analyses conducted for the needs of the present paper also revealed that patients who considered epilepsy to be the cause of sex life difficulties obtained lower scores in the aspect of the quality of life in all the subscales of the QOLIE-31 questionnaire. Furthermore, such patients more commonly declared dissatisfaction with their sex life and difficulties with sleep initiation. The results might be due to the belief shared by the respondents that they have an irreparable defect related to the disease, which makes them useless in numerous aspects (also in terms of sexual attractiveness), which may undermine their satisfaction with life and even the meaning of life. It is possible that the respondents were characterized by a specific style of the attribution of the causes of sexual life problems. Subjects may cognitively locate the causes of their failures within themselves, so they are certain that they are unable to experience satisfaction in this area of life [36,37]. Heersink and his associates described the mechanism of the so-called attribution error in the explanation of the nocebo effect. The tendency of the subject to regard the preference of external or internal sources of information concerning one’s emotional status is important. It is influenced by various personality-related variables, such as focusing on oneself, increased sensitivity to one’s emotional status, predisposition to introspection, sensitivity to social anxiety, and concentration on information from the environment [37]. Undoubtedly, it is worth considering the cognitive representation of the disease and the patient’s beliefs concerning the effects of the disease and its treatment. It requires an individualized approach and comprising contexts in which the specific beliefs of the patient are dominant.

Issues concerning the sexual area and sleep disorders should constitute an integral part of medical history during follow-up specialist visits, especially because the manifestations tend to exacerbate over the course of the disease [26]. Henning et al. reported that only 16% of the participants were asked about sexual disorders during medical consultations [16]. Appropriate patient education in this respect should be performed at the early stage after diagnosing the disease [38]. Sleep disorders occur in over half (57.1%) of children with epilepsy [39]. It was demonstrated that the implementation of a suitable therapy in childhood improves their quality of life [40].

More attention has recently been given to mindfulness-based cognitive therapy to improve the quality of sex life in patients with epilepsy via the reduction in the perception of stress [41]. Sleep disorders may be improved with the use of sleep monitoring and education in terms of sleep hygiene [42]. The significance of the interdisciplinary approach to the treatment of anxiety and depression in patients with epilepsy was also emphasized as a factor improving the quality of life [43]. Physical activity is another form of therapy in the case of sleep disorders in children. Its effect was demonstrated in a study conducted in children with epilepsy [44].

It was also attempted to treat sleep disorders with pharmacotherapy regimens. A study conducted by Ayala-Guerrero demonstrated a positive influence of gabapentin on sleep disorders in patients with epilepsy [45]. The use of perampanel increased the total hours of sleep, sleep quality and the time of awakening [46]. In the case of drug-resistant epilepsy, the use of clobazam also reduced the frequency of seizures and improved sleep quality. Patients experienced fewer depressive symptoms and reached better scores in the assessment of the quality of life [47]. In the case of sexual disorders, newer generations of drugs, including lamotrigine, oxcarbazepine and levetiracetam, had a less marked influence on the sexuality of patients [20]. Operative treatment is proposed in order to improve the general quality of life in patients with drug-resistant epilepsy and severe psychiatric manifestations [48].

### 4.1. Limitations of the Study

The study was conducted on a low number of participants. Therefore, the results need to be viewed with caution. The presence of problems associated with sexual functioning and sleep initiation problems was not confirmed with objective methods. It was also not clearly specified how the respondents understood the term “sexual satisfaction”. We did not identify a wide range of factors (biological, mental, and psychosocial), which may be related to problems with sleep initiation and/or problems with sexual functioning.

### 4.2. Strengths of the Study

The study was conducted in several centers in Poland, which facilitated the collection of information from patients living in various environments. The results indicate the directions and necessity of conducting further research in this area. The realization of the present study indirectly provided the authors with the possibility of confirming their belief that the issues tackled should not be neglected in clinical settings because patients were eager to continue conversations concerning their sexual functioning and/or sleep.

## 5. Conclusions

Over half of the respondents declared problems with sleep initiation, and less than half declared being definitely satisfied with their sex life. The respondents claimed that epilepsy contributed to difficulties with their sex life. Patient declarations concerning sexual functioning and sleep initiation problems revealed a correlation with almost all the subscales of the QOLIE-31 questionnaire.

The authors emphasized the important aspects of patient functioning, which are rarely within the scope of interest of clinicians. It was demonstrated that the interest in the complaints and problems of patients regarding sexual function and problems with sleep initiation (even at the level of self-assessment) and treating them as symptoms might indicate the need for further diagnostics and treatment. The issues should constitute the constant element of medical history taking, as improvement in these areas of functioning may promote better quality of life in patients with epilepsy.

## Figures and Tables

**Figure 1 brainsci-11-00778-f001:**
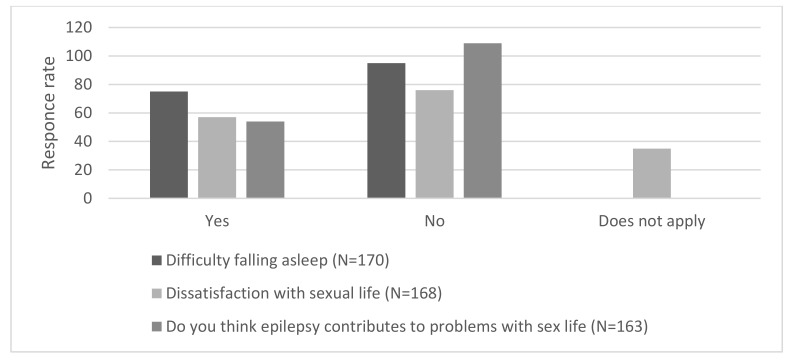
Frequency of satisfaction with sex life, the quality of sleep and the opinion of the influence of epilepsy on the quality of sex life and the subject’s self-esteem.

**Table 1 brainsci-11-00778-t001:** Sociodemographic data.

Data		N = Number of Subjects	%
Marital status	Married Unmarried Divorced/separated Widowed Domestic partnership	75 68 15 4 7	44.4% 40.2% 8.9% 2.4% 4.1%
Number of Children	0 1 2 3 4 5	80 41 29 12 3 1	48.2% 24.7% 17.5% 7.2% 1.8% 0.6%
Education	Tertiary Secondary Vocational Primary	28 84 38 20	16.5% 49.4% 22.4% 11.8%
Place of Residence	Village Town < 10,000 inhabit. Big town > 10,000 < 100,000 inhabit. City > 100,000 inhabit.	52 27 44 46	30.6% 15.9% 25.9% 27.1%

**Table 2 brainsci-11-00778-t002:** Type and frequency of seizures.

Type of Seizure	Generalized Tonic-Clonic Seizures		Complex Partial Seizures		Simple Partial Seizures		Absence Seizures		Myoclonic Seizures		Unclassified Seizures	
Frequency of Seizures	C	%	C	%	C	%	C	%	C	%	C	%
No seizures over the past 6 months	53	50	32	32.3	17	48.6	8	57.1	3	33.3	3	1.8
1–2 seizures over the past 6 months	29	27.4	23	23.2	6	17.1	2	14.3				
3–5 seizures over the past 6 months	10	9.4	10	10.1	3	8.6	1	7.1	1	11.1		
1 or more seizures per month	8	7.5	21	21.2	5	14.3						
1 or more seizures per week	6	5.7	12	12.1	4	11.4	2	14.3	2	22.2		
1 or more seizures per day			1	1.0			1	7.1	3	33.3		

^1^ C—the number of cases in the specified category and type of seizure, %—the respective percentage concerning the type of seizure.

**Table 3 brainsci-11-00778-t003:** QOLIE-31 results in the groups reporting and denying sleep initiation difficulties, a comparative analysis (statistically significant correlations *p* ≤ 0.001 in bold).

	Sleep Initiation Problems	N	Average	Standard Deviation	t	*p*
SW	yes	75	41.27	24.40	**−5.181**	**0.0001**
	no	95	61.77	26.45		
QQ	yes	75	48.50	22.18	**−2.836**	**0.005**
	no	95	58.96	25.75		
EWB	yes	75	40.51	6.52	**−2.140**	**0.034**
	no	95	42.47	5.07		
EF	yes	75	41.20	8.63	−0.687	0.493
	no	95	42.07	7.88		
COG	yes	75	32.73	13,46	**−4.820**	**0.0001**
	no	95	43,25	14.58		
ME	yes	75	45.11	26.65	**−3.616**	**0.0001**
	no	95	61.23	30.40		
SF	yes	75	47.37	20.78	**−3.939**	**0.0001**
	no	95	60.28	21.46		
QOLIE-31	yes	75	41.25	11.30	**−5.175**	**0.0001**
	no	95	50.79	12.36		

(SW—seizure worry, OQ—overall quality of life, EWB—emotional well-being, EF—energy and fatigue issues, COG—cognitive functioning, ME—medication effects, SF—social functioning).

**Table 4 brainsci-11-00778-t004:** QOLIE-31 results in the groups reporting satisfaction and dissatisfaction with sex life, a comparative analysis (statistically significant correlations *p* ≤ 0.001 in bold).

	Satisfaction with Sex Life	N	Average	Standard Deviation	t	*p*
SW	Yes	76	57.64	29.60	**2.27**	**0.024**
	No	57	46.24	26.03		
QQ	yes	76	60.60	25.04	2.83	0.005
	no	57	48.46	23.51		
EWB	yes	76	41.76	5.02	1.56	0.121
	no	57	40.25	5.87		
EF	yes	76	43.77	7.67	**4.08**	**0.0001**
	no	57	38.38	7.30		
COG	yes	76	42.20	13.92	**2.24**	**0.027**
	no	57	36.31	16.26		
ME	yes	76	51.85	31.08	−1.35	0.179
	no	57	59.11	30.00		
SF	yes	76	59.52	23.29	**2.62**	**0.010**
	no	57	49.58	20.20		
QOLIE-31	yes	76	50.06	13.00	**3.08**	**0.003**
	no	57	43.12	12.01		

(SW—seizure worry, OQ—overall quality of life, EWB—emotional well-being, EF—energy and fatigue issues, COG—cognitive functioning, ME—medication effects, SF—social functioning).

**Table 5 brainsci-11-00778-t005:** QOLIE-31 results in the groups declaring that epilepsy was vs. was not the cause of problems with sex life, a comparative analysis (statistically significant correlations *p* ≤ 0.001 in bold).

	Epilepsy as the Cause of Problems with Sex Life	N	Average	Standard Deviation	t	*p*
SW	yes	54	39.08	25.23	**−4.591**	**0.0001**
	no	109	59.10	26.61		
QQ	yes	54	45.37	23.77	**−3.397**	**0.001**
	no	109	58.87	23.86		
EWB	yes	54	39.63	6.33	**−2.617**	**0.010**
	no	109	42.22	5.10		
EF	yes	54	38.80	7.08	**−3.300**	**0.001**
	no	109	43.19	8.41		
COG	yes	54	31.85	13.31	**−4.301**	**0.0001**
	no	109	41.95	14.46		
ME	yes	54	46.40	29.69	**−2.394**	**0.018**
	no	109	58.20	29.54		
SF	yes	54	44.61	20.66	**−4.403**	**0.0001**
	no	109	59.85	20.83		
QOLIE-31	yes	54	39.44	10.98	**−5.498**	**0.0001**
	no	109	50.13	11.99		

(SW—seizure worry, OQ—overall quality of life, EWB—emotional well-being, EF—energy and fatigue issues, COG—cognitive functioning, ME—medication effects, SF—social functioning).

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
