# Peer review of "The Quality of Life in Patients with Epilepsy in the Context of Sleep Problems and Sexual Satisfaction"

_brainsci, 2021, doi:10.3390/brainsci11060778_

Round 1

Reviewer 1 Report

The manuscript presents a cross-sectional analysis of the quality of life in patients with epilepsy in the context of sleep problems and sexual satisfaction. The variables addressed in the study are generally well supported by the Introductory part of the study. However, several references that are currently included in the Discussion section would rather fit the Introduction part. 

After reading the manuscript and the ways the authors have assessed the sexual satisfaction in relation to the other two dimensions (QoL and sleep problems), I do consider that the study is rather an exploratory one and I kindly advise the authors  to state about the exploratory nature in the study, i.e. in the title and in the body of the manuscript.

This particular study has, in my point of view, the value of a needs' analysis to further explore the sexual satisfaction in relation to QoL (and other variables) of patients diagnosed with epilepsy. It would help in the future to include information about the existent diagnosed sexual dysfunctions of the participants, as well as to use instruments (existent scales and/or questionnaires) for the assessment of the components of sexual satisfaction and sexual functioning. The usage of only one question on sexual satisfaction offers a singular subjective perception of the variable, and the answer might be biased by social desirability and/or by the lack of understanding of the concept (the last is one of the limits of the study which was pointed out by the authors).

The Results section is rather descriptive one and it is not clear why the authors did generate the two categories of participants in regards to the self-perceived sexual satisfaction, instead of using the points of the Likert scale as continuous variable in their statistical analysis involving the other variables, e.g. scores of the sub-scales of the general score of the QoL instrument, the age of the participants, the marital status etc. For example, regression analyses would offer some valuable insights.

Another aspect which is not clear in the Results section: Did the authors perform an analysis of the normal distribution of their data? Are the criteria met for performing the t test analysis? 

Several interactions among variables (age, gender, marital status) might be considered in the Results section.

As already indicated above, the Discussion part includes a rich body of references, which would fit into the Introduction section. I do recommend to the authors to focus more on the interpretation of their own results in the Discussion part and refer to other studies in case that those are supporting specifically their own findings. 

Reviewer 2 Report

The authors investigated sleep problems and sexual satisfaction in patients with epilepsy. For this purpose, a questionnaire was conducted, in which about 200 people took part. The authors' main conclusion is that respondents claimed that epilepsy contributed to difficulties with their sex life. This conclusion is quite trivial.

Although the topic of quality of life in epilepsy patients is significant, the article suffers from methodological inaccuracies. Because of this, the authors' results are difficult to interpret.

Major issues.

The sex and age structure of the sample is unclear. The authors indicate that the mean age was 40 years, the range was from 18 to 84 years. Obviously, sleep problems and sexual life satisfaction will vary among people of different ages, so it is not appropriate to combine people of all ages into one group.

The specifics of the diagnosis, how often epileptic seizures are observed, and whether sleep and sex life patterns are related are unclear. In addition, it remains unclear whether these disorders are specific to epileptic patients or are also characteristic of healthy subjects of the same sex and age cohort.

It is unclear why the authors chose to combine sleep and sex life rather than any other determinants of quality of life in this article. The reference to Hippocrates does not seem convincing "The significance of sleep disturbance in the context of sexual disorders was emphasized by Hippocrates in the 4th century BC"

The authors somewhat arbitrarily classify responses from the standardized questionnaire into two groups. For example, they classify the answer "sometimes" as YES in one case and NO in the other (page 4). How did this affect the standardized questionnaire? The questionnaire should probably be revalidated in this case.

Round 2

Reviewer 1 Report

The authors have addressed the comments from the reviewers in an adequate manner and information was added accordingly.

In my opinion, the manuscript is now suitable in content and presentation to be accepted for publication in this journal. 

Author Response

Dear Reviewer,

Thank you for all valuable remarks. It was a pleasure to cooperate with you.

Best regards,

Kornelia Zareba and Co-authors

Reviewer 2 Report

The authors mostly addressed my comments. 

Minor comments:

1) lines 223-225. It is necessary to specify what statistical criteria were used and describe them in the Methods. Please, include in the text the values of the statistical tests used.

2) Figure 1 should be changed because according to Methods (lines 189-202), other response categories were used in the data analysis.
